# Torsional Stability Assessment of Columns Using Photometry and FEM

**Krzysztof Wierzbicki**, **Piotr Szewczyk \***, **Wiesław Paczkowski**, **Tomasz Wróblewski and Szymon Skibicki**

Faculty of Civil Engineering and Architecture, West Pomeranian University of Technology in Szczecin, 70-311 Szczecin, Poland; Krzysztof.Wierzbicki@zut.edu.pl (K.W.); Wieslaw.Paczkowski@zut.edu.pl (W.P.); Tomasz.Wroblewski@zut.edu.pl (T.W.); Szymon.Skibicki@zut.edu.pl (S.S.)

\* Correspondence: szewczyk@zut.edu.pl

**Abstract:** This paper presents a numerical analysis of the load-carrying capacity of steel open-section columns of a coal power plant structure. The structure was subjected to soil subsidence, which led to considerable structural deformations and damages. As a result, additional stresses appeared in the structure, and the static scheme of the structure was changed. To assess the influence of structural changes on the safety of the structure, a detailed investigation was necessary. Laser scanning was used to collect information concerning the geometry of structural elements. Results of the scanning were implemented in a numerical model of the structure. A complex finite element method (FEM) shell model of the column in ABAQUS software was developed. Torsional buckling stability analysis of column members was carried out. Different boundary conditions depending on the type of column connections to other elements were considered. Torsional deformations were treated as imperfections. Analysis showed that the connections of bracing elements, e.g., beams in multilevel frame, directly affected the collapse mechanism and load-bearing capacity of the investigated element. Finally, the paper showed that an appropriate change in the connections between the analyzed column and multilevel frame beams prevents the column from twisting, thereby increasing the critical force and load-bearing capacity of the analyzed industrial structure.

**Keywords:** torsional buckling; point cloud; critical load; FEM

## 1. Introduction

Access to modern technologies has dramatically changed the work of engineers today. Work with already existing structures is a branch of civil engineering that requires more effort from the engineer than building new objects. All types of renovation, modernization, structural strengthening, or changes in the way objects are used require thorough analysis of a given structure's performance before, during, and after implemented changes, as well as verification of the existing technical specification, which frequently needs to be written from scratch.

Buckling of main elements is the one of the most important problems which should be considered in structural strengthening. Deformations of old and inappropriately designed structures can be the main reason for buckling [1]. For complex structures, introducing structural strengthening requires performing advanced numerical analyses, which consider different loading scenarios in time (static and dynamic), environmental conditions [2,3], and changes in the materials properties in time [4,5]. Currently, there are many interesting strengthening systems for damaged structures that fulfil their role even in extreme environmental conditions [6].

To date, any survey concerning the state of structure relied on the tedious work of people conducting successive measurements, which was time-consuming and generated costs. Laser scanning

technology is a new tool whose usefulness cannot be overestimated. It enables precise measurement and is capable of generating complete point cloud data. Introduction of laser scanning was a revolution that changed the approach of how to work with existing structures. The technique has a wide range of applications in many sectors of industry and science [7,8], including civil engineering [9–11]. The laser scanning point cloud technique was used to analyze a complex, multistory, steel industrial structure in the presented paper.

Another class of tools that engineers have at their disposal is a range of software used to design, prepare, and execute construction operations. The group includes finite element method (FEM) programs that enable static and dynamic analysis. These are applications optimized toward particular branches of engineering, as well as professional, scientific programs with an almost unlimited scope of analytical potential, e.g., ABAQUS. This paper shows how these two groups of programs complement each other in the analysis of complex engineering problems.

Effective analysis of a complex structure must inevitably lead to certain simplifications that enable developing and analyzing a model in limited time. On the other hand, simplifications mean that some phenomena are disregarded. Torsional instability is one of those cases. Although the problem has been known for a long time, documents setting current standards [12] seem to marginalize it so that it can be easily overlooked. Therefore, this paper pays special attention to torsional forms of instability in the analyzed structures.

## 2. Research Issues of the Structure

### 2.1. Description of the Structure

Analysis was conducted on a steel frame industrial structure made up of four identical segments with expansion joints. Each segment was 72 m long, 91 m wide, and 54.3 m high. The arrangement of structural axes of the main part of the segment is presented in Figure 1. The columns of the load-bearing structure had flanged cruciform sections and were welded from steel plates. The seating joints of the columns were also made by welding horizontal sheet metal elements 50 mm thick. These are accompanied by transoms and lateral and horizontal bracing made of rolled beams, built-up beams, and trussed elements. The main technological levels were placed at the height of +10.5 and 25.5 m with a reinforced concrete slab resting on a rectangular grid of steel floor beams. Additional levels were made at +34.5 and 45 m with steel grids filled with platform gratings.

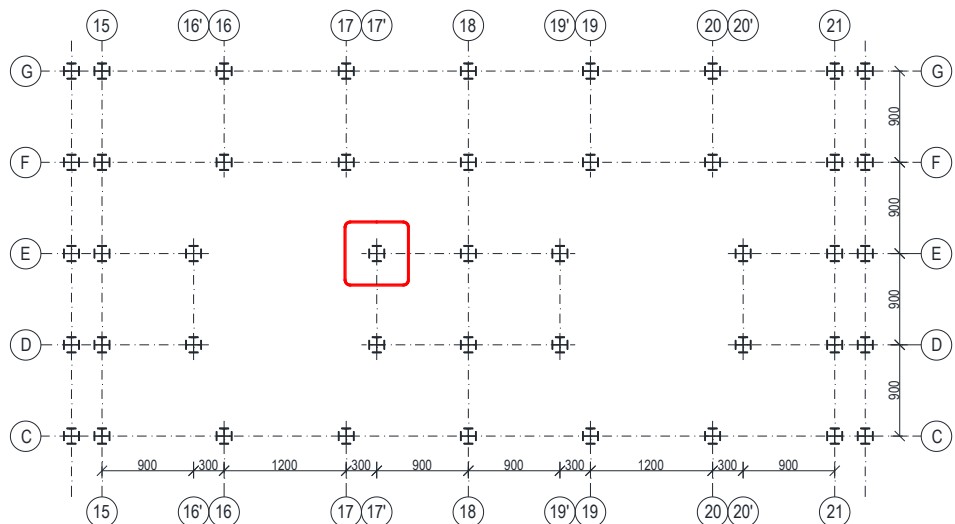

**Figure 1.** Scheme of structural axes of one segment of analyzed object.

The column marked in Figure 1 with a red frame is one of four elements that carry the load of a boiler with a mass of 3700 t.

The structure was made of carbon steel with a design yield strength of 215 MPa. In line with standards valid during its construction [13], it was St3S steel with three levels of oxygen reduction: rimmed, semi-killed, and killed. Killed steel used then can be compared to currently produced S235JR [14] steel.

### 2.2. Conducted Repair Works and Used Technologies

The structure in question was analyzed by researchers many times [15–17]; many evaluations were written about its technical condition, and many projects of how to strengthen it were put forward. This was due to sudden and uneven subsidence of the ground surface over 15 years ago. Although subsidence was slowed down owing to some repair works, it has continued to proceed at the rate of several millimeters per year. Currently, the maximum value of vertical subsidence at the floor level exceeds 200 mm. This kind of nonstatic loading causes dangerous limiting stress. Some bars have become plastic. Some bracings have buckled under compressive load, and some have broken under tension load.

The structural columns have already been strengthened, some even twice. All strengthening works so far extended the cross-sectional area. This was due to the fact that axial force plays a dominant role in columns and that the determined buckling coefficient for compression was slightly smaller than one (approximately 0.95). The risk of losing flexural stability is, therefore, marginal. However, all analyses so far used bar models. This approach is generally correct, given the scale of the object. Nevertheless, bar models enable monitoring of phenomena linked to warping of the section, which can lead to torsional loss of stability. That is why one column was selected as best representing other elements (marked in Figures 1 and 2) for further, thorough analysis with spatial shell models. The models enabled analysis of the column's behavior under axial forces, bending moment, and clearly observed imperfections that resulted from subsidence and mistakes made during assembly.

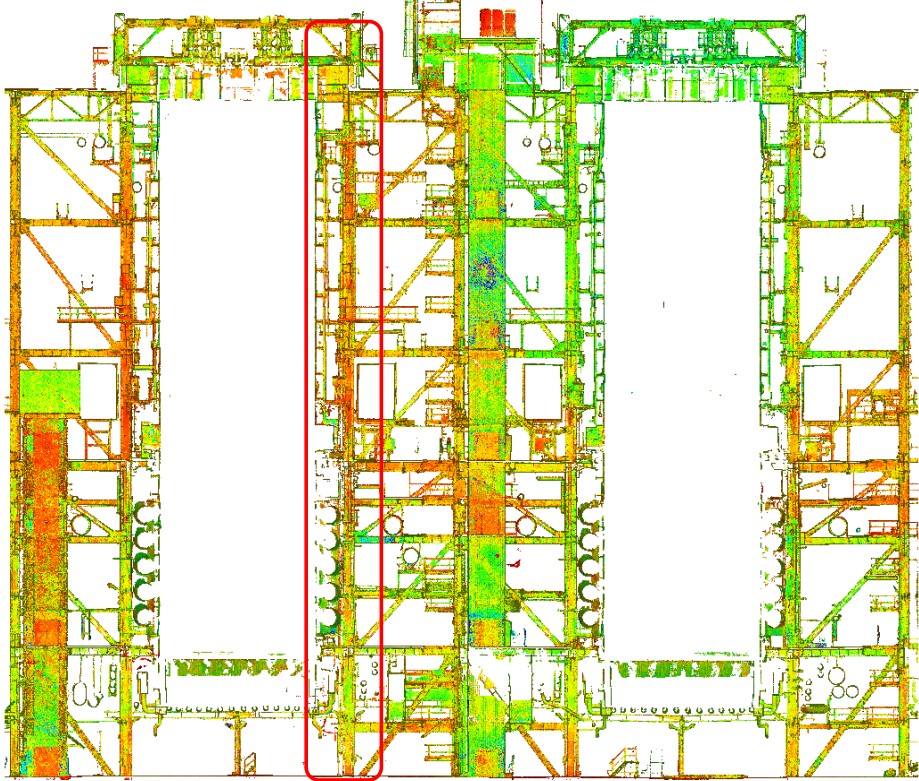

**Figure 2.** An example of point cloud section in vicinity of E axis.

The scale of the structure, many previously conducted renovations, difficult access to structural elements resulting from the height of the building, and high temperature in the vicinity of the industrial installation inside it made its complete survey a difficult task. Therefore, the structure was three-dimensionally (3D) scanned to produce a cloud of points (Figure 2) that could later be more easily processed and analyzed. Measurements were made with a Leica ScanStation P40(Leica, Wetzlar, Germany) scanner with a linear accuracy of 1.5 mm + 10 ppm (parts per million), which, given the structure's size, is precision not available to other methods. The obtained point cloud was a great source of data. It was used to compare the present morphology with the design documentation, to analyze damaged or ruptured elements, or to look into instability cases. Point cloud analysis was also more effective as data, once collected, did not require frequent visits to the building where the production process could run uninterrupted.

## 3. Torsional Buckling

### 3.1. State of the Art

The foundations of lateral–torsional buckling (LTB) were laid in the first half of the 20th century. Vlasov [18] formulated the general form of the static equilibrium differential equation, and his contribution was to include properties of thin-walled members under torsional load. To calculate critical force for any given beam under compressive force, one has to account for lateral buckling of the element in two planes perpendicular to each other (which should be crossing through the main axes of the cross-section) and torsional buckling. It can be given by the following system of coupled differential equations [19]:

$$
\begin{aligned}
E \cdot I_y \cdot w^{IV} - N \cdot (w' - y_s \cdot \varphi') &= 0, \\
E \cdot I_z \cdot v^{IV} - N \cdot (v' - z_s \cdot \varphi') &= 0, \\
E \cdot I_\omega \cdot \varphi^{IV} - G \cdot I_t \cdot \varphi^{II} + N \cdot (- y_s \cdot w^{II} + z_s \cdot v^{II} - i_s^2 \cdot \varphi^{II}) &= 0,
\end{aligned}
\tag{1}
$$

where E is Young's modulus, G is the shear modulus, $I_y$, $I_z$ are moments of inertia, $I_\omega$ is the warping moment of inertia, $I_t$ is the torsional moment of inertia, N is the axial load, w, v are translations of the center of gravity after deformation (Figure 3), $y_s$, $z_s$ are coordinates of shear center according to the center of gravity (Figure 3), $\varphi$ is the angle of rotation about the longitudinal axis of an element (Figure 3), $i_s = y_s^2 + z_s^2 + (I_y + I_z)/A$, and A is the area of the cross-section.

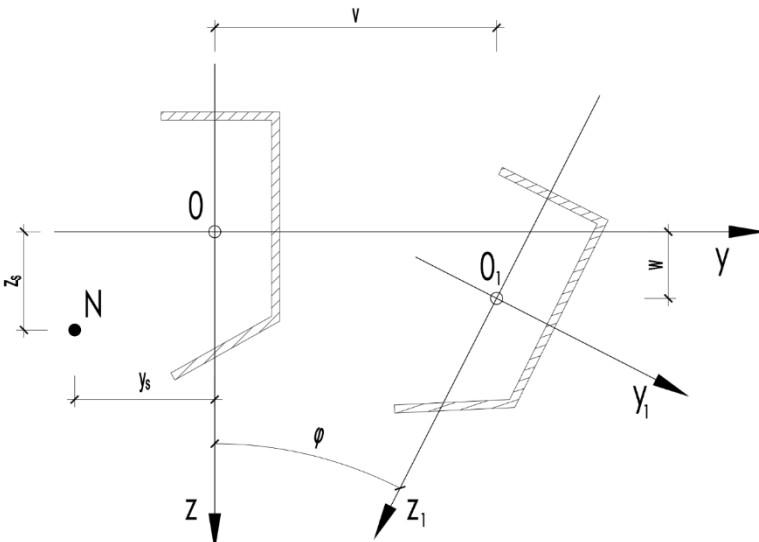

**Figure 3.** Current and reference cross-section of an open profile according to Equation (1).

When there are no lateral or rotational supports, and when an element has the possibility of torsional and flexural buckling in two perpendicular planes, the equilibrium state can be written in one equation as follows:

$$(N_{cr,y} - N) \cdot (N_{cr,z} - N) \cdot (N_{cr,T} - N) \cdot i_s{}^2 - y_s \cdot \alpha_{yw}J^2 \cdot (N_{cr,z} - N) - z_s{}^2 \cdot \alpha_{zw}J^2 \cdot (N_{cr,y} - N) = 0, \quad (2)$$

where $N_{cr,y}$, $N_{cr,z}$ are critical forces of flexural buckling in two planes, which are perpendicular to each other, $N_{cr,T}$ is the torsional critical force, and $\alpha_{cw}$, $\alpha_{zw}$ are buckling coefficients related to boundary and load conditions, derived from Vlasov [12] and Brezina [20] equations.

Using approximate equations, the critical force for a pinned I-beam, which is not transverse-loaded and has no warping supports at both ends, can be derived as follows [13]:

$$\begin{aligned} N_{cr,y} &= \pi^2 \cdot E \cdot I_y/L^2, \\ N_{cr,z} &= \pi^2 \cdot E \cdot I_z/L^2, \\ N_{cr,T} &= (\pi^2 \cdot E \cdot I_\omega/L^2 + G \cdot I_t)/i_s{}^2, \end{aligned} \quad (3)$$

where L is the buckling length.

### 3.2. Standard Conditions

The current standard of design for steel structures [12] only sets a condition of torsional load-bearing capacity for members not sensitive to cross-sectional distortion. Total torsional moment in any given cross-section is determined as the sum of free torsional moment (of St. Venant) and lateral–torsional moment [12]. In Section 6.3 of the standard [12], stability of members is only checked in determination of lateral buckling capacity for compressed elements and for flexural–torsional buckling capacity of elements bent or simultaneously bent and compressed. Please note that the standard [12] does not provide information on how to determine bending critical moment ($M_{cr}$), which is necessary to find a relative slenderness ratio for lateral–torsional buckling to ultimately determine flexural bending capacity accounting for LTB. Interestingly, the necessary equation can be found in a standard for aluminum structures [21]. Section 6.3.1.4 of the standard [12] assumes that, in the determination of relative slenderness ratio, the critical force resulting from torsional buckling is greater than critical force of lateral or lateral–torsional buckling. In other words, the assumption says that torsional instability is not going to happen because the member loses stability due to other factors.

Compressive capacity accounting for lateral buckling and bending capacity accounting for lateral–torsional buckling are given by the following equations:

$$\begin{aligned} N_{b,Rd} &= \chi \cdot A \cdot f_y/\gamma_{M1}, \\ M_{b,Rd} &= \chi_{LT} \cdot W_y \cdot f_y/\gamma_{M1}, \end{aligned} \quad (4)$$

where $\chi$ is the flexural buckling coefficient, $\chi_{LT}$ is the lateral–torsional buckling coefficient, $f_y$ is the yield strength, and $\gamma_{M1}$ is the partial safety factor for resistance of a member to buckling.

To protect a beam against torsional failure, the following condition of the standard [12] in Section BB.2.2 must be fulfilled:

$$C_{\vartheta,k} = K_\vartheta \cdot K_\upsilon \cdot M_{pl,k}{}^2/(EI_z), \quad (5)$$

where $C_{\vartheta,k}$ is the rotational stiffness per length (e.g., for sandwich panels and T-sheets), $K_\upsilon = 0.35$ for elastic analysis, $K_\upsilon = 1.00$ for plastic analysis, $K_\vartheta$ is the partial factor due to bending moments diagram and boundary conditions (see Table BB.1 [12]), and $M_{pl,k}$ is the characteristic value of plastic resistance of bending for a cross-section.

### 3.3. Effect of Torsional Bracing on Load-Bearing Capacity

As suggested in the previous section, design of steel structures should be made so that loss of stability in compressed elements due to lateral or flexural–torsional buckling happens before torsional

buckling. Compressed elements lose stability when they are under relatively large normal force and when they are characterized by a higher relative slenderness ratio regarding flexural buckling than regarding torsional buckling. This is the case when bracings can limit translation of an element but cannot protect it from torsion.

To increase the torsional capacity of a member, you can brace it using dedicated bracings against torsion or you can strengthen it by enlarging existing stiffening plates (gilts or membranes). The most commonly used plates are warping braces which connect the upper and lower flanges of an I-section. Warping braces are commonly used in beams for other structural reasons. Endplates (Figure 4a) can connect a steel column with the foundation. They make it possible to use a column to support a girder or to connect wall or ceiling girts with a girder. It is sensible to think about the use of warping braces at the stage of designing steel structures. Bimoment bracing (Figure 4b) is another type of strengthening which connects the upper and bottom flanges (without the web). The planes of sheet metal are parallel to the web of the element that is being strengthened. There are other strengthening methods (see [22]) using modified X-shaped batten plates and closed profile stiffeners which can be found in the literature. However, as these methods are not the subject matter of the paper, they are not discussed at length.

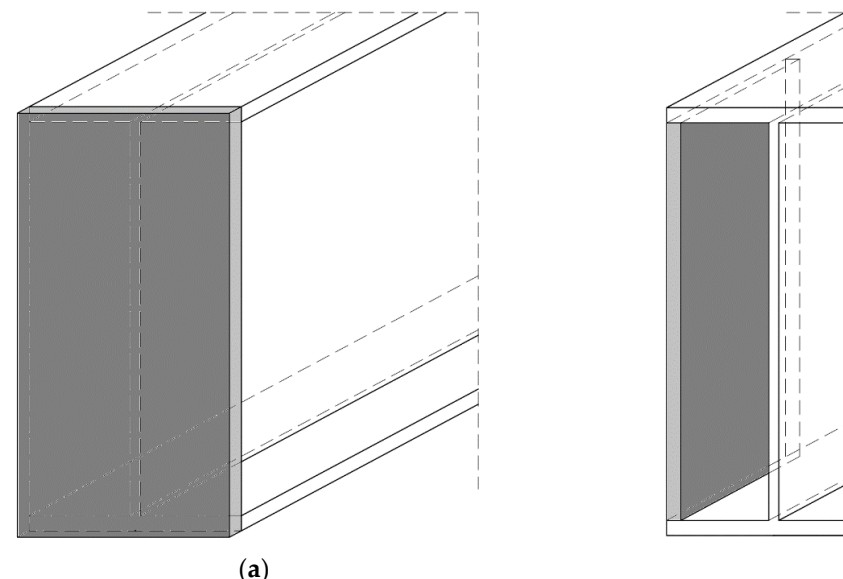
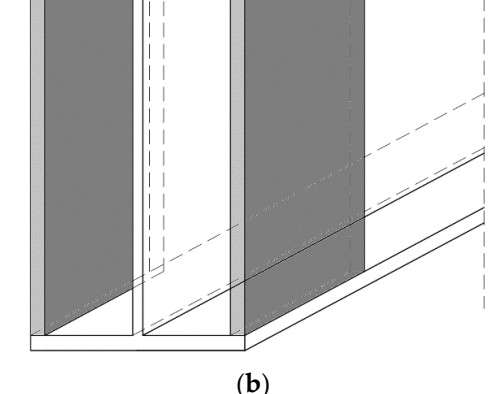

(**a**)  (**b**)

**Figure 4.** Torsional restraint bracing using stiffening metal sheets: (**a**) endplates; (**b**) bimoment bracing.

Torsional bracing prevents the section from warping, which affects the displacement and internal forces of nonfree torsion. Bimoment bracing seems to be the most effective method [22,23]. It uses metal sheets parallel to the web or other elements. Bimoment bracing systems have high torsional strength, which prevents sections from warping. Endplates and web stiffeners have the smallest impact on preventing profile warping. They must use sheet metal of large thickness (e.g., over 30 mm for IPE300 L = 5000 mm beam [24]) to achieve adequate torsional stiffness, which could limit profile warping and, thus, increase the load-bearing capacity of the element.

Elastic supports that prevent warping are used in calculations using the energy method [19]. Fundamental function coefficients are approximated to polynomial functions as follows:

$$\Delta\Pi = \Delta U_{s,1} + \Delta U_{s,2} - \Delta T, \tag{6}$$

where $\Delta\Pi$ is the overall energy, $\Delta U_{s,1}$ is the elastic energy in a torsional–flexural state, $\Delta U_{s,2}$ is the elastic energy of restraints, and $\Delta T$ is the work made by an external load.



For a pinned I-beam under an evenly distributed load with warping restraints at supports, Equation (6) can be derived as follows [25]:

$$\Delta U_{s,1} = 0.5 \cdot (E \cdot I_z \cdot \int d^2u/dx^2)^2 dx + G \cdot I_t \cdot \int (d\delta/dx)^2 dx + E \cdot I_\omega \cdot \int (d^2\delta/dx^2)^2 dx),$$
$$\Delta U_{s,2} = 0.5 \cdot \alpha_w \cdot ((d\delta/dx)^2{}_{x=0} + (d\delta/dx)^2{}_{x=L}),$$
$$\Delta T = 0.5 \cdot q_z \cdot (\int \delta d^2u/dx^2 \cdot (L-x)xdx + z_g \cdot \int \delta^2 dx),$$

(7)

where u is the translation in a plane perpendicular to the plane of bending, $\delta$ is the angle of rotation between the plane of bending and the plane of the web of a deformed beam, $\alpha_w$ is the elastic stiffness of a warping restraint, $q_z$ is the magnitude of an evenly distributed load, $z_g$ is the coordinate of load according to the center of gravity (negative for a destabilizing load and positive for a stabilizing load).

## 4. Finite Element Study

### 4.1. Finite Element Model

A numerical model of the investigated column was developed in the ABAQUS/CAE 2018 environment on the basis of a 3D scanning point cloud. It is a 3D model that accounts for geometric and material nonlinearity. Flanges and webs were modeled using shell elements with reduced integration and linear shape functions (S4R). The model was recalculated using shell elements with full integration and a linear shape function (S4); there was no significant difference between outcomes, but the calculations lasted longer. Therefore, S4R elements were adopted for further analyses. Steel constituents were modeled using a bilinear, elastic–plastic model with stiffening. The yield strength limit ($f_y$ = 215 MPa) was kept in line with the standard valid when the structure was designed [13].

The cross-section of the column (Figure 5) was verified with the point cloud. Owing to the high accuracy of measurements made at many points of the structure (Figure 5b), it was even possible to determine the thickness of steel elements where access is very difficult, e.g., the web.

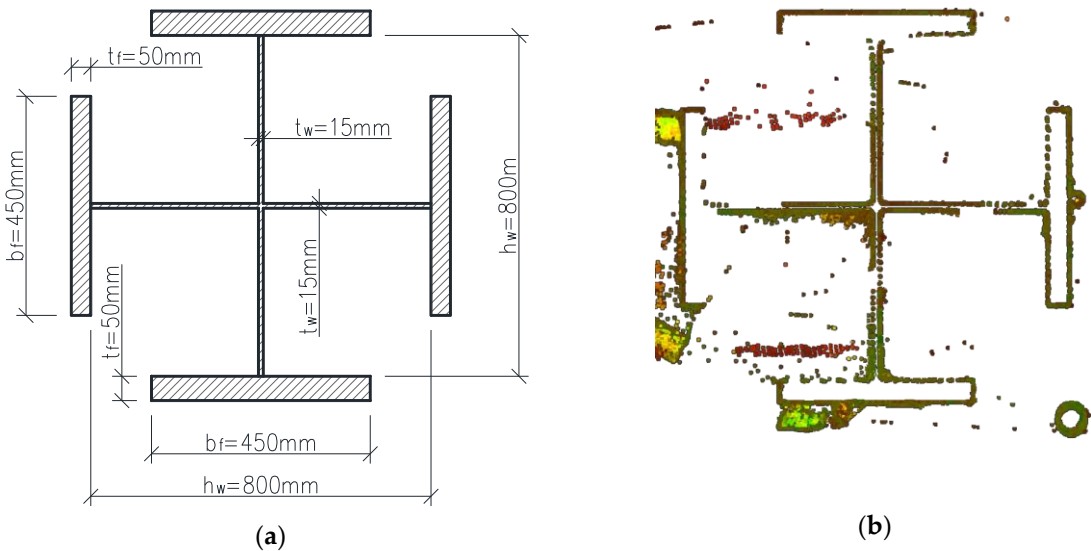

**Figure 5.** Cross-section of the column: (**a**) geometrical dimensions; (**b**) point cloud view.

Because shell models were used in the study, it was important to use correct boundary conditions that properly reflected real parameters of how flanges and webs were mounted on the column. It was specified in the previous section that lower levels of the structure, up to +25.5 m, had a massive reinforced concrete floor resting on steel beams mounted to the column. At higher levels, the access

to critical elements of the structure is only provided by steel platform gratings, which significantly changes the column's performance. Therefore, two calculation models were developed.

The first model covers the lowest level from the foundations to the first floor at the height of +10.5 m. The base plate is fastened in reinforced concrete (Figure 6a), over 150 cm from its top surface. This massive anchoring was modeled through blocking all three translational degrees of freedom on all the column edges (Figure 6b).

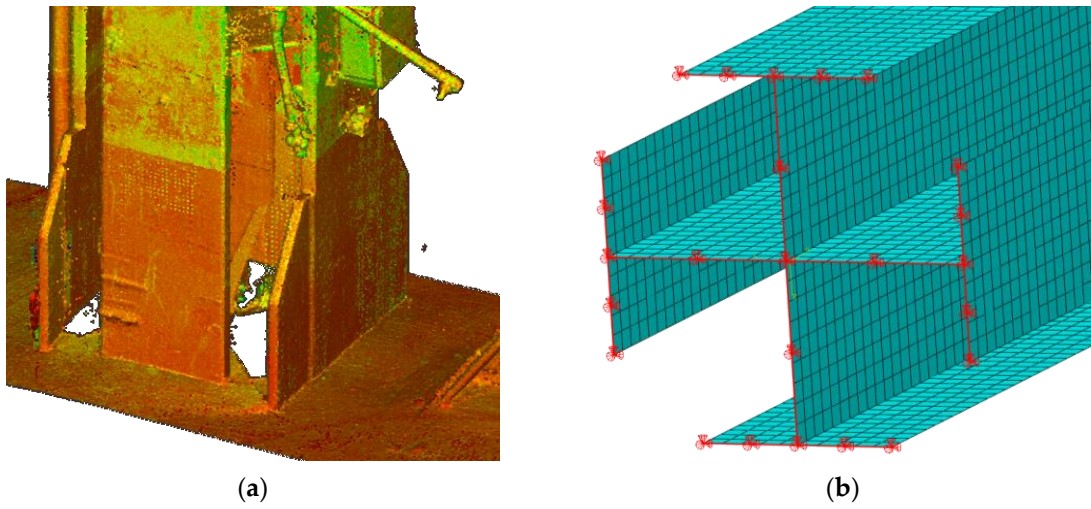

| (**a**) | (**b**) |

**Figure 6.** Boundary conditions at the height of 0 m: (**a**) point cloud; (**b**) numerical model.

Steel I-beams are connected perpendicularly to the column at the height of 10.5 m. The beams support reinforced concrete floor with a thickness of 150 mm (Figure 7a). Because the upper flanges of the beams carry the weight of the floor, their twisting is impossible. This successfully protects the column from rotation at this level. To map these support conditions in the numerical model, the top of the column was modeled as a nondeformable slab, the so-called rigid body, connected to the flanges and the web (Figure 7b). The center of the element was designated as a reference point with the following boundary conditions: first, two translational, parallel degrees of freedom in the slab plane were blocked which prohibited horizontal translation of the column's upper flange; second, one rotational degree of freedom was blocked, which blocked the column's rotation along its axis, i.e., prevented column twisting. Linear displacement along the column's axis was not blocked.

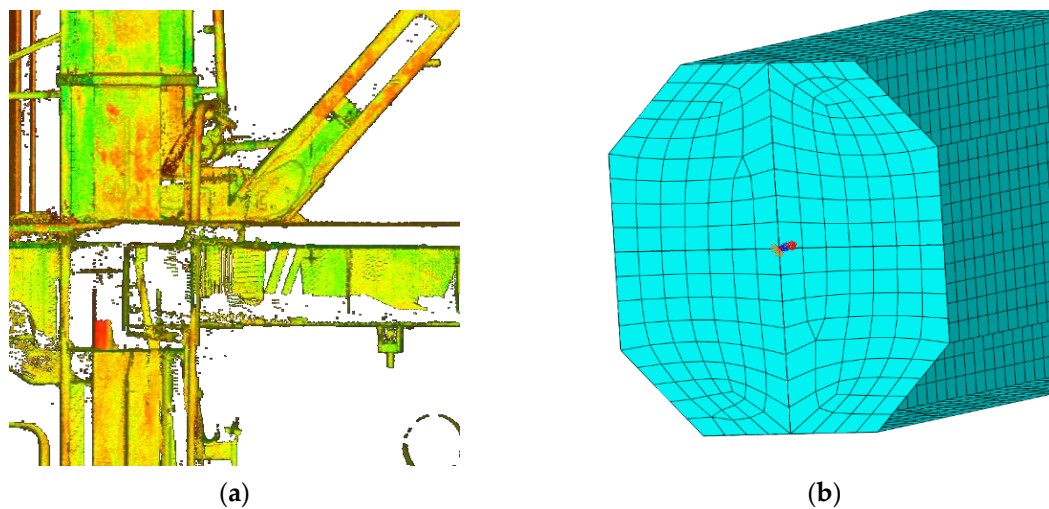

| (**a**) | (**b**) |

**Figure 7.** Boundary conditions at the height of 10.5 m: (**a**) point cloud; (**b**) numerical model.

The second model covers the column from the height of +25.5 m, i.e., the level of the last reinforced concrete floor where the column's rotation along its own axis was blocked, up to its head at +52.3 m. The boundary conditions at +25.5 m were the same at those at the base (see Figure 6b).

The head of the column at +52.3 m consists of a horizontal steel plate (thickness of 50 mm) and a bearing providing linear support for the industrial installation and boiler. The bearing allows free rotation in one direction, which can clearly be seen in Figure 8a. The numerical model introduced a sheet metal plate with the same thickness as in the real structure. A linear support was added to the model in the direction of the bearing that blocked translational degrees of freedom in the plane of the column's head and rotational degrees of freedom blocking the head's rotation along the column's axis (Figure 8b).

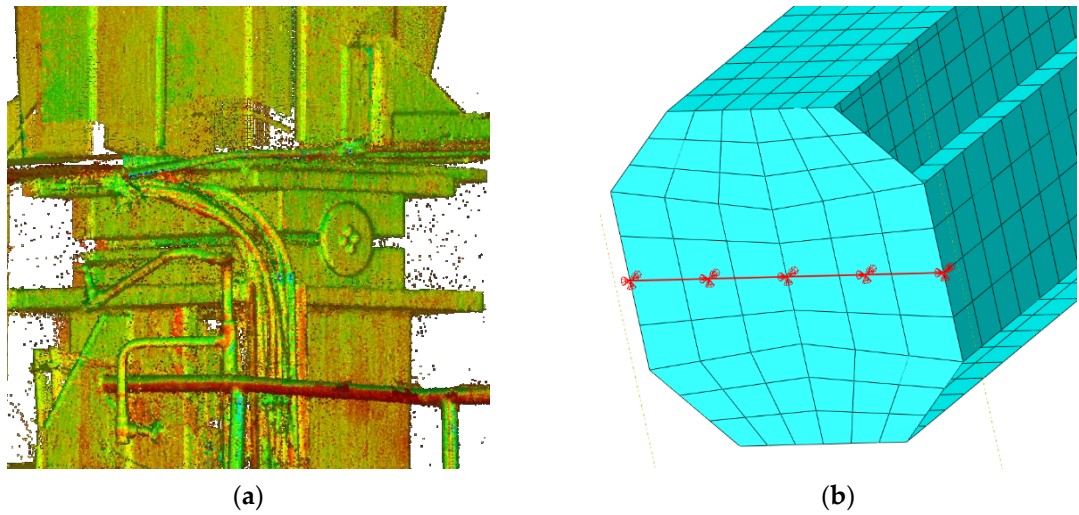

(**a**)  (**b**)

**Figure 8.** Boundary conditions at the height of 52.3 m: (**a**) point cloud; (**b**) numerical model.

Between the extreme floors at +25.5 m and 52.3 m, there are two additional floors at +34.5 m and +45 m where horizontal beams are connected to the column. Unquestionably, these connections reduce the buckling length of the column under flexural buckling. However, regarding torsional stability, it is necessary to consider in detail how the horizontal beam is connected to the column. The method of connecting beam flanges with column flanges is particularly important. To this end, the 3D scan of the structure was thoroughly analyzed. The scan turned out to be very useful as it provided data about connections located very high or in places with limited access. On the basis of point cloud analysis, three ways of connecting horizontal beams with the column were discerned:

1. welding connection;
2. bolted joint with endplate;
3. shear connection with bolts covering only the web.

With regard to torsional susceptibility, it was assumed that beams connected using the first two methods would be considered in the analysis as elements limiting the twisting movement of the column's cross-section. Elements connected with the third method were disregarded in the analysis, owing to the fact that the shear connection covering only the web provided little possibility of preventing rotation. Figures 9 and 10 present point clouds representing the mounting conditions of successive horizontal beams connected to the column at levels without reinforced concrete slabs.

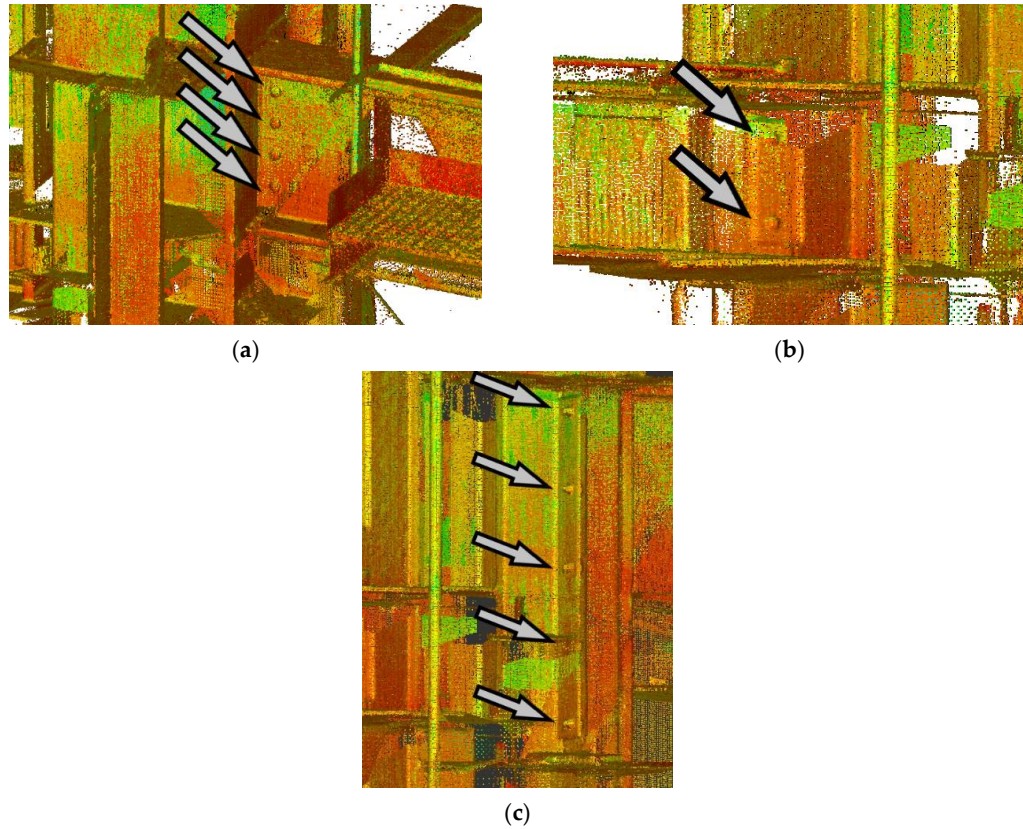

**Figure 9.** Level +34.5 m with indicated connection bolts using arrows: (**a**) view of axis 17′ in direction of row F; (**b**) view of axis 17′ in direction of row D; (**c**) view of row E in direction of axis 18.

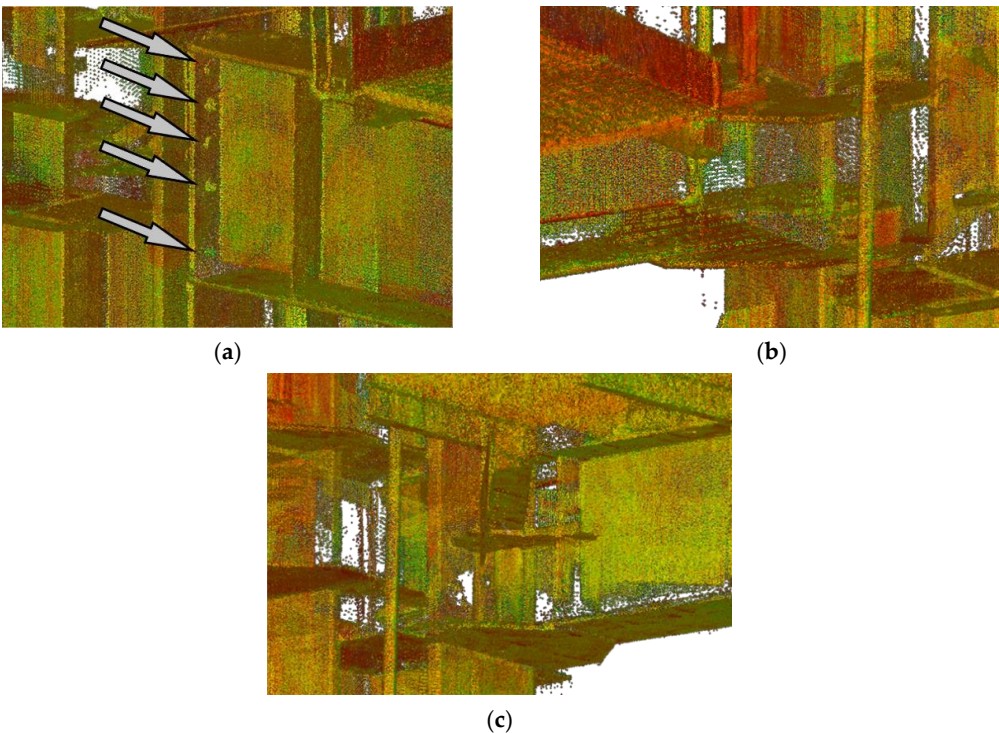

**Figure 10.** Level +45 m with indicated connection bolts using arrows: (**a**) view of axis 17′ in direction of row F; (**b**) view of axis 17′ in direction of row D; (**c**) view of row E in direction of axis 18.

On the basis of the above assumptions, only two beams at +45.5 m visible in Figure 9a,b were initially taken into account. The model used beams of the real length, and the boundary conditions at their ends reflected their real mounting method in adjacent columns. A view of the model is presented in Figure 11a. For the further analysis below, additional models were also developed which accounted for all horizontal beams touching the column (see Figure 11b).

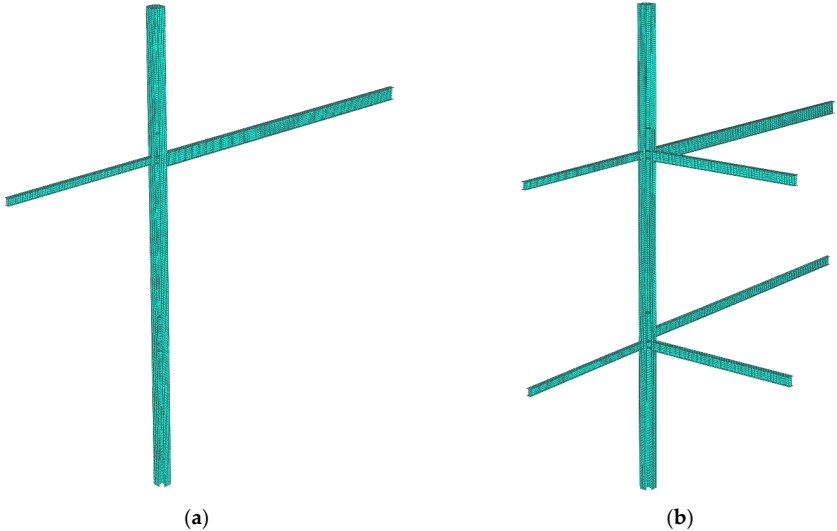

|     |     |
| :-: | :-: |
| (**a**) | (**b**) |

**Figure 11.** Numerical model of the column: (**a**) supported with beams with stiff connections; (**b**) supported with all beams.

### 4.2. Eigenvector Evaluation

ABAQUS enables determination of the critical force and stability loss form. Critical force is determined for an ideal elastic material, i.e., below a certain slenderness limit, linked to the yield point. It is, therefore, a theoretical value which does not have a real equivalent. Nevertheless, it is used in standard algorithms [12,13] for the determination of load-bearing capacity of members at risk of losing stability. In our analysis, stability loss determination was more useful (see examples in Figure 12).

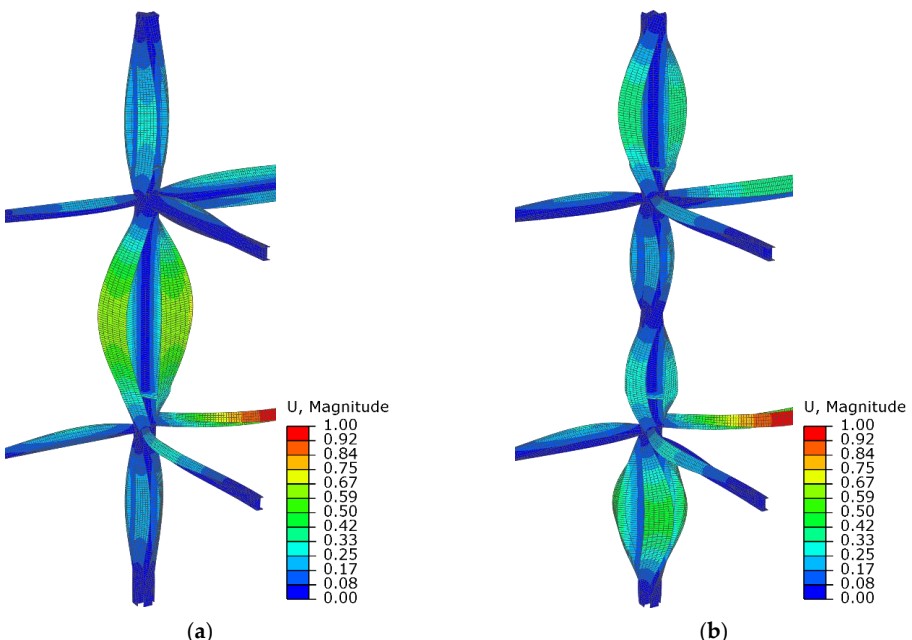

|     |     |
| :-: | :-: |
| (**a**) | (**b**) |

**Figure 12.** Stability loss of upper part of the column: (**a**) first mode; (**b**) second mode.

### 4.3. Model of the Column with Initial Imperfection

From the practical point of view, the first buckling mode, linked to the lowest critical force, is the most important one. The first mode was, therefore, used to model initial geometrical imperfection of the column. ABAQUS enables recording translations of points collected in buckling analysis and then changing the geometry of the initial model of other types of analysis, e.g., static nonlinear analysis. Thus, it is possible to develop a geometrical model with initial imperfection. The user has the freedom to scale imperfection to fit their needs. Note that the program treats the deformed model as the starting point of analysis. That is why the translation and stress values of the initially translated points of the finite element mesh are zero. Figure 13 shows examples of nonlinear models with initial torsional imperfection consistent with the first mode of stability loss.

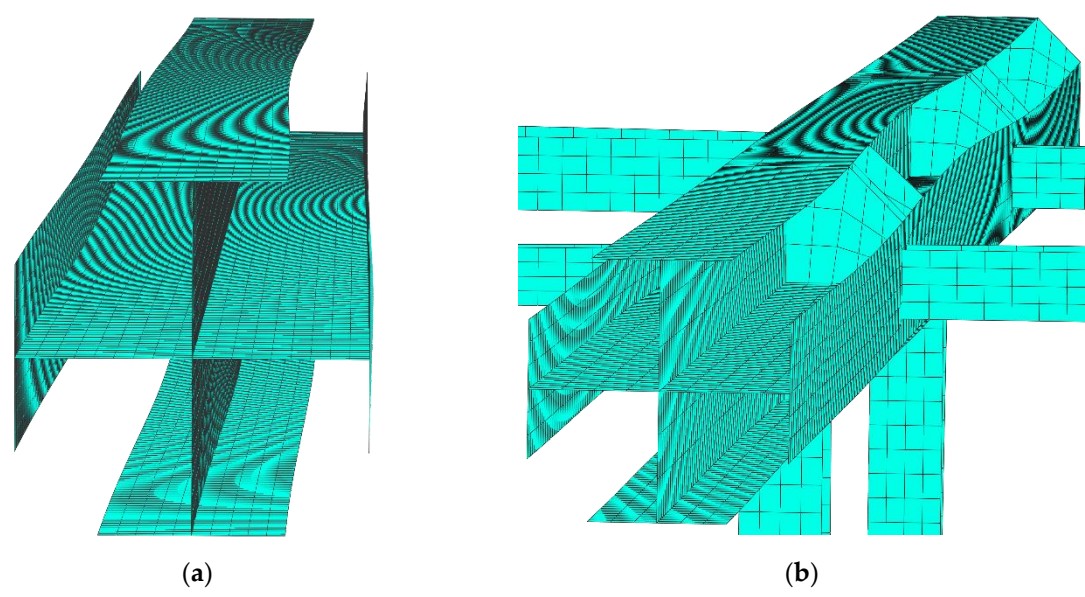

(**a**) (**b**)

**Figure 13.** Model with initial imperfection (scale factor = 1.0 for geometrical imperfection L/250): (**a**) lower part 0–10.5 m; (**b**) upper part 25.5–52.3 m.

A 3D scanning shape verification of the existing reference column confirmed that its deformation was comparable to the first buckling mode of its idealized equivalent.

Having the geometrical shape of the existing structure (geometry determined from the point cloud enables finding distances between respective points of the element), it was possible to determine the imperfection amplitude implemented in ABAQUS. Naturally, the initially deformed geometry of the numerical model deviates from the real member as it is based on the idealized buckling mode. The approximation was assumed to be satisfactory because an "ideal" shape of imperfection would produce a lower critical force and, thus, stress values that would be on the "safe side".

We did not know loads acting on the column during 3D scanning. Likewise, we did not know how the ground deformation or applied load affected real deformation. In the analysis of other columns, an assumption was made that the amplitude of existing deformation would be fully taken into account as imperfection amplitude in the direction of the first buckling mode. The assumption can generate a lower load-bearing capacity and, therefore, produces a safe result. The analysis of calculated results showed that, up to the point of stability loss, additional twisting had a lower value than the real value measured with the real geometry. It revealed that most existing torsional deformations in the real column were caused by uneven subsidence of the ground and not by applied load. This finding confirmed the correctness of the assumptions made.

Deformation amplitude had different values relative to the location of a given column. For a given column, deformation amplitude of torsional imperfection was approximately equal to 1/250 of the

length of the column span between lateral stiffening (Figure 14). The assumed torsional amplitude was four times greater than that recommended in the literature (L/1000) [26], which scales the shape of imperfection consistent with the first mode of lateral–torsional buckling for elements being bent with warping.

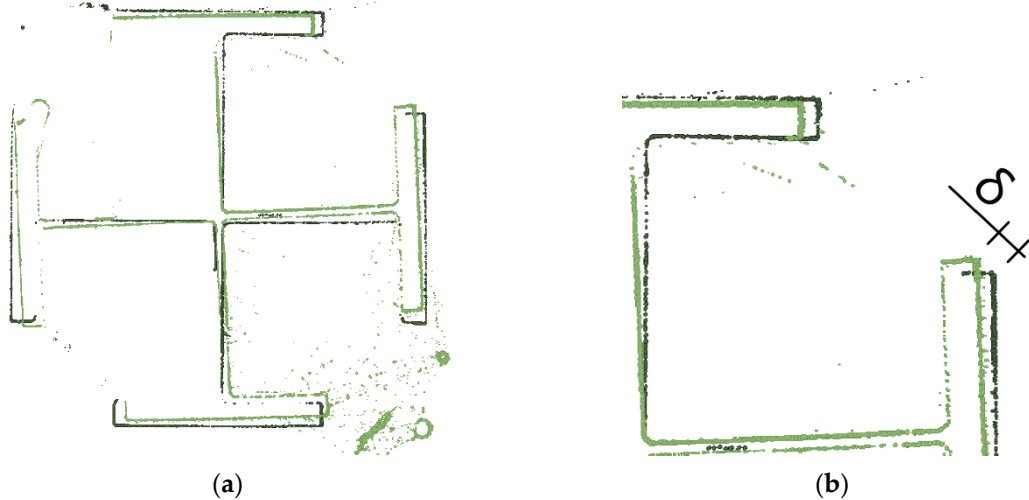

(**a**)                                                                                                (**b**)

**Figure 14.** The shape of torsional deformation of real structure. The section marked in black represents the section shape at the ends of the column. The section marked in green was located near the center of the height of the column: (**a**) Complete cross sections; (**b**) The magnitude of torsional imperfection read from the real structure. δ ≈ L/250, where L is the length of the column segment.

The current standard for dimensioning of steel elements [12] provides an alternative method of taking imperfection into account. The shape of imperfection can be assumed to be convergent with the $\eta_{cr}$ mode of elastic buckling, and imperfection can be determined from the following formula:

$$\eta_{cr} = e_0 \cdot (N_{cr}/(E \cdot I \cdot \eta_{cr,max}) \cdot \eta_{cr} = e_0/\lambda^2 \cdot (N_{Rk}/(E \cdot I \cdot \eta_{cr,max}) \cdot \eta_{cr}, \tag{8}$$

where $\eta_{cr}$ is the deformation due to flexural buckling, $\lambda$ is the relative slenderness, $e_0$ is the initial imperfection derived from (5.10) [12], $N_{Rk}$ is the compressive resistance of a cross-section ($N_{pl,Rk}$), and $E \cdot I \cdot \eta_{cr,max}$ is the bending moment in a critical cross-section from ηcr.

## 5. Results and Discussion

### 5.1. Column at the Height of 0–10.5 m

In the first phase of analysis, the critical force relative to its loss of stability was determined. In our case, the critical force was 125 MN and significantly exceeded the ultimate limit state (ULS) of the cross-section, which was slightly over 24 MN. This means that damage would occur in the plastic regime of deformation. Figure 15a shows the first, torsional mode of stability loss.

In the next step, the initially deformed geometry of the column was superimposed on the model used for plastic analysis of the performance of the element. Figure 15b shows the stress map according to the von Mises hypothesis for a damaged element. One can see characteristic plastic areas on one side of each flange in the twisted column. The model was loaded according to the most adverse combination of loads for the column. Axial force, which plays a dominant role in this element, was approximately 20.2 MN. The Riks method was used in calculations. The load was increased gradually until its maximum.

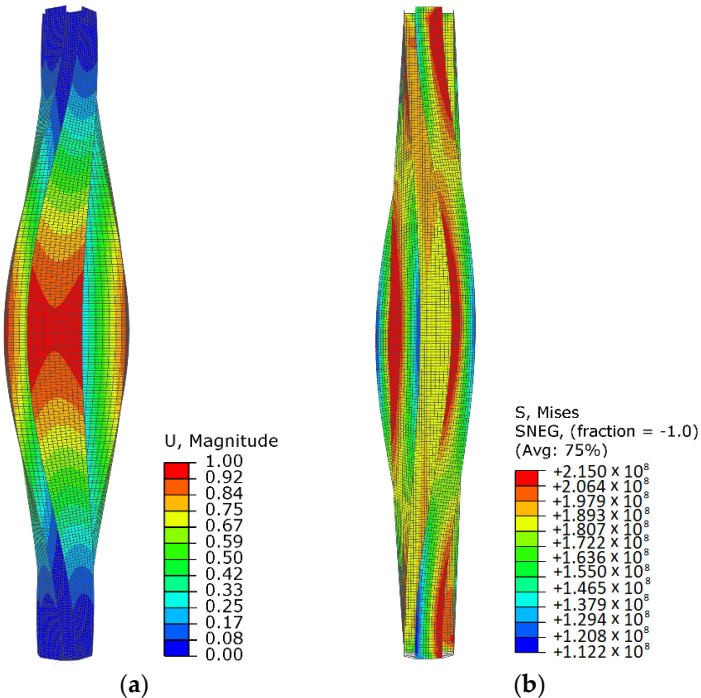

(**a**)    (**b**)

**Figure 15.** Bottom part of the column: (**a**) mode of stability loss; (**b**) von Mises stress within damage range.

Results of the relationship between axial force and displacement are presented in Figure 16. Displacement is understood to be the biggest horizontal translation along the column's length in one of its nodes. Figure 16 shows four lines. The red line is the point of reference and shows the behavior of an ideal model without initial imperfection. One can see a long linear range and sudden destruction in the final phase. Insignificant displacements in the linear range are due to bending moment impact. The green and blue lines represent models with L/500 and L/250 initial imperfection, respectively. The black dotted line shows axial force resulting from the most adverse combination of loads. Figure 15 clearly shows the impact of initial imperfection on horizontal displacement and the force/displacement relationship. In the model closer to reality, the linear range of operation is significantly shorter, and the transition to a dangerous regime, tantamount to damage, is smoother. Note that models with initial imperfection had lower serviceability limit states.

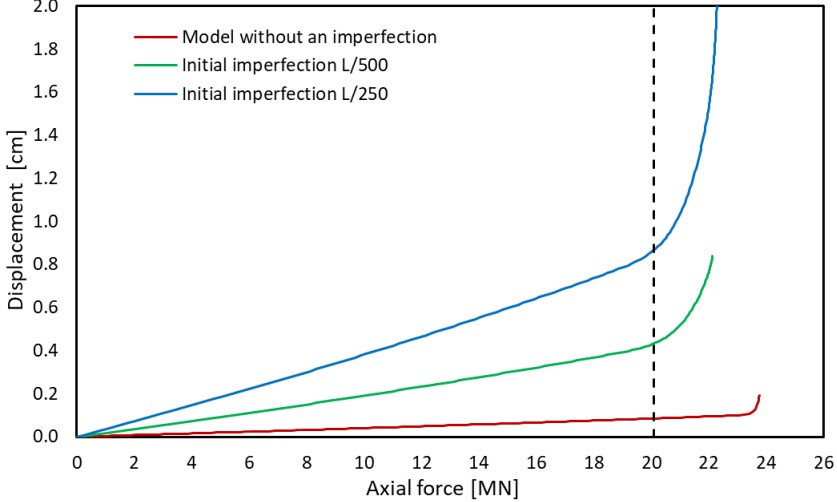

**Figure 16.** Force/displacement relationship for models in the range of 0–10.5 m.

### 5.2. Column at the Height of 25.5–52.3 m

Analysis of this part of the column also started from a determination of the critical force and buckling mode of the model presented in Figure 11a. The L/250 initial imperfection was used for all analyses in this section. The assumption was that only beams connected stiffly with the column (through welding or endplates) work together. Such elements could, in reality, limit warping of the column's cross-section. For this assumption, the first mode of stability loss was flexural with an equivalent force of 48 MN and the second mode was torsional with a slightly greater force of 52 MN. The next step was nonlinear analysis that determined the ultimate limit state (ULS) of the column with the initial imperfection. Figure 17 shows the axial force/horizontal displacement relationship represented with a blue continuous line.

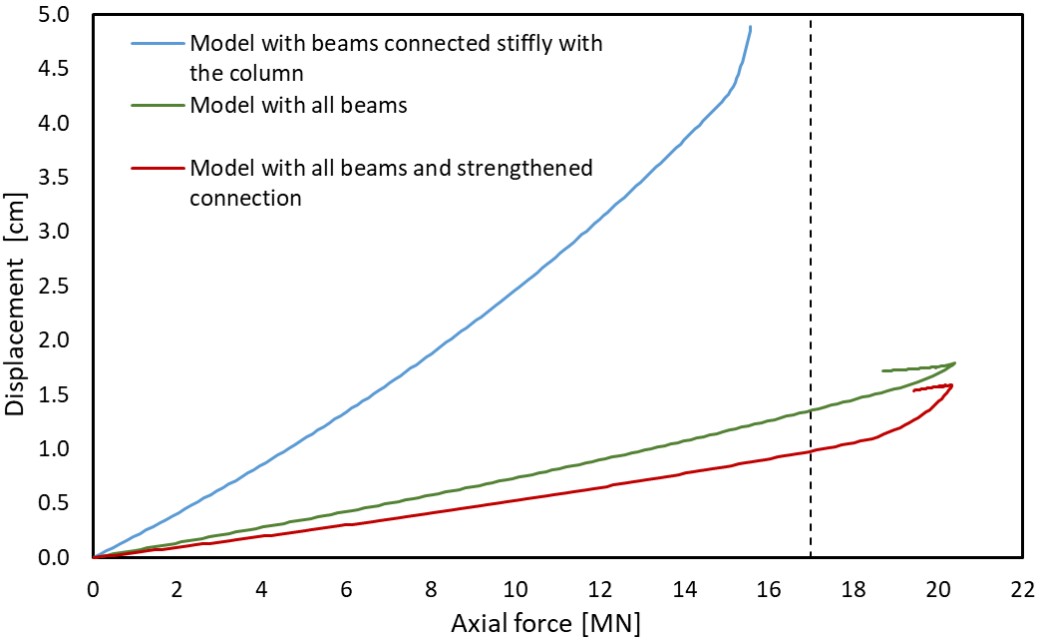

**Figure 17.** Axial force/displacement relationship for models in the range of 25.5–52.3 m.

Note that maximum axial force for the most adverse combination of loads was approximately 17 MN, i.e., it exceeded the ULS. To increase the ULS of the column, it was necessary to reduce its buckling length and limit the risk of cross-sectional warping. Therefore, a model closer to reality was developed in the next step. That model contained all six horizontal beams connected to the column. Some of them were linked to the column only through webs, thus modeling the shear connection with bolts covering only the web of a beam. These connections can be seen in Figures 9 and 10. Figure 18 shows the FEM model with connections that cover only the webs.

The small effect of web stiffness on blocking the column's rotation caused a slight increase in critical force up to 57.2 MN. In this case, the first mode of stability loss was torsional. Nonlinear analysis results are shown in Figure 17 with the green line. Despite the slight increase in critical force, the limitation of displacement along the column's length resulted in a substantial increase in ULS and, thus, to partial plasticizing of the most stressed cross-sections.

The final step in analysis was to develop a way of increasing the strength of the column and other elements against stability loss. As mentioned above in the paper, a heavy reinforced concrete slab resting on a steel grid was a natural element that limited displacement and warping. There were no slabs at the higher levels. Many earlier analyses [27–29] concluded that introduction of such heavy elements that rest on column tops could limit warping, particularly in conditions when torsional stability loss is possible. However, the authors of this paper put forward another solution. In our

opinion, the most effective solution would be to fully use the stiffness of beams that are connected to the column. In many cases, beams rest on seats made of welded plates, which can be clearly seen in Figure 9b,c and Figure 10b,c. Unfortunately, analysis of the point cloud does not provide data on whether or not there are any (and if so in what condition) joints connecting beam flanges with plates of the supports. A systematic review of existing connections and complementing missing joints seems to be a relatively simple solution. More importantly, it does not introduce too many welding deformations to the structure.

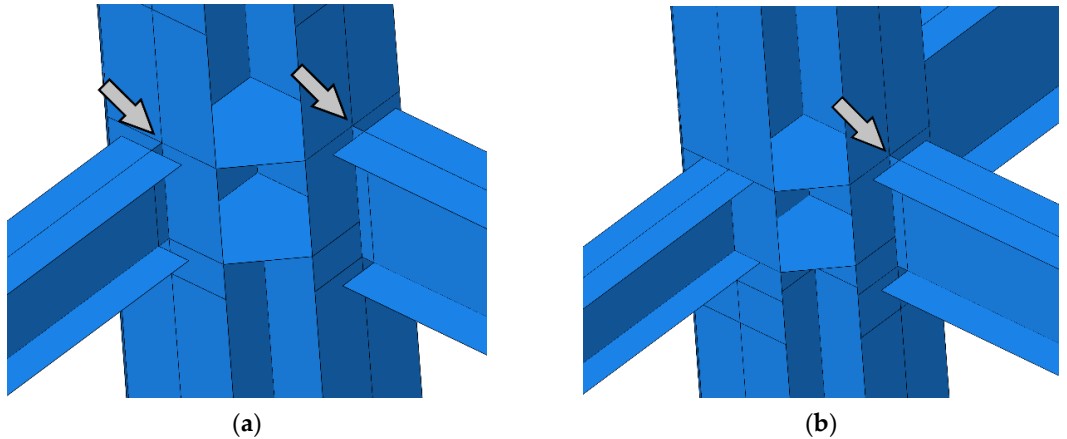

(**a**)                                             (**b**)

**Figure 18.** Model showing overlap beam connections: (**a**) +34.5 m level; (**b**) +45 m level.

The assumption of full stiffness of all six beams connected with the column at levels +34.5 and +45 m results in a significant increase in critical force up to 88.8 MN and causes torsional stability loss, as presented in Figure 19a.

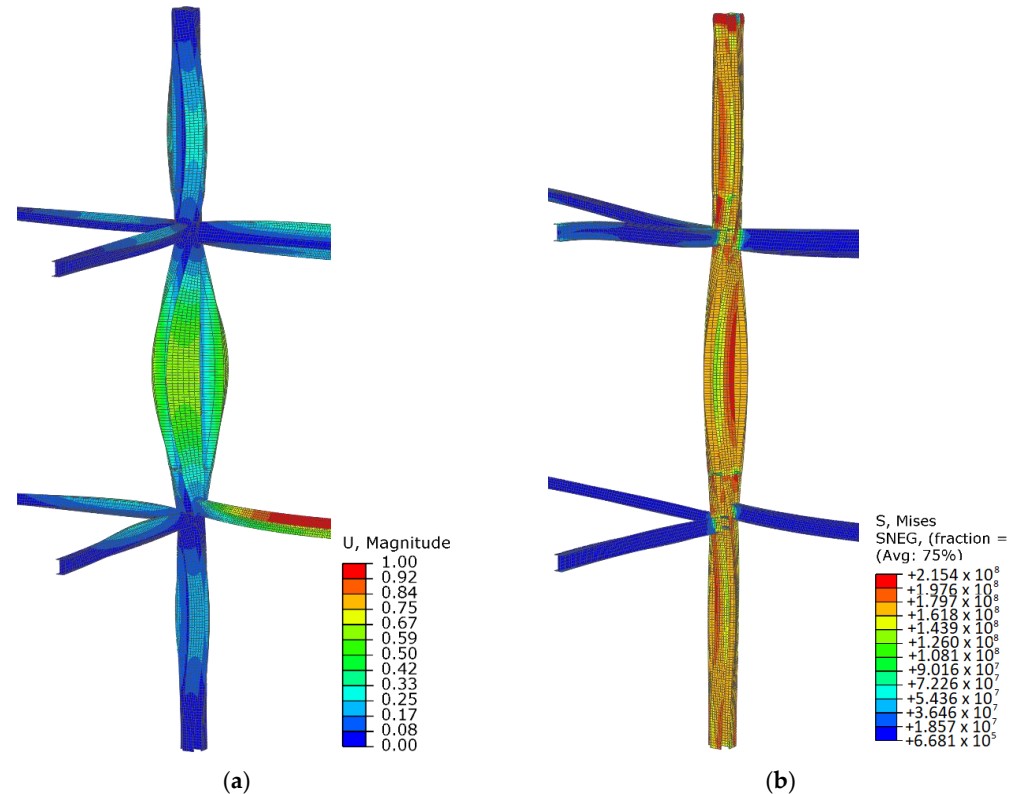

(**a**)                                             (**b**)

**Figure 19.** Upper part of the column: (**a**) stability loss mode; (**b**) von Mises stress in damage regime.

The presented mode of stability loss was used, as done previously, as a form of initial imperfection in nonlinear analysis. An imperfection amplitude of approximately L/250 was read from the geometry of the point cloud. The obtained result is presented in Figure 17 as axial force/horizontal displacement relationship (the red line). A significant increase in the element's critical force due to full use of the stiffness of beams, which up to that point were connected only through their webs, did not result in an increase in ULS, as the analysis accounted for the nonlinear character of the steel. However, stiffness of the whole column increased, which was shown as lower horizontal displacement. Figure 19b shows a stress map according to the von Mises hypothesis in the damage regime. One can see stress areas exceeding the yield point, forming patterns common for torsion.

## 6. Conclusions

This paper presented calculation results for models of the same column using different approaches to mapping boundary conditions. On the basis of the point cloud analysis, the information necessary to model initial imperfections was obtained. This information would be impossible to obtain using any other methods of structure survey/diagnostics. This was an attempt to make a theoretical computational model more consistent with the actual performance of the member. Finally, the paper proposed a simple method of strengthening the already existing structure. To increase the column's load-bearing capacity, it is recommended to provide all horizontal beams connected to columns with joints linking their flanges with those of the columns. It can be done as follows: first, by joining flanges with seats, which already exist in many places; second, via introduction of additional elements, e.g., overlays joining beam and column flanges. Owing to a large number of design options chosen for the analyzed structure, individual solutions are recommended for each individual node of the building.

This paper attempted to address the issue which is often disregarded in designing new and analyzing old structures. Beam (bar) elements with open profiles are susceptible to torsional stability loss. This is particularly important when dealing with a compressed element of small lateral slenderness (i.e., relative slenderness over 0.9), where bracings limit displacement and simultaneously, to a smaller degree, prevent the column from twisting. Correct estimation of the load-bearing capacity of members undergoing torsional buckling is often problematic due to difficulty with mapping boundary conditions and, thus, correct determination of buckling length. One should also be aware that calculation with beam elements does not even provide the possibility of correctly mapping boundary conditions, with regard to torsion. This is why an element that is being dimensioned is protected against torsion. This is much easier when designing new structures, but it can be difficult and expensive for already existing structures that need strengthening. The point cloud analysis confirmed the appearance of twisted columns in the given structure. This confirms the thesis that open sections are susceptible to torsional stability loss.

The most important conclusion from the present study is the realization that the method of connecting stiffening beams (boundary conditions) has a large impact on the load-bearing capacity and stability of the analyzed column. Reinforced concrete floors resting on upper flanges of horizontal beams mounted to the column significantly increase the torsional stiffness of these connections. Places where the floors are missing leave much room for analysis on the effect of flexible connection stiffness on the member's serviceability limit state.

**Author Contributions:** Conceptualization, K.W., P.S., W.P., T.W., and S.S.; methodology, K.W., P.S., and W.P.; software, K.W. and P.S.; validation, K.W. and P.S.; formal analysis, K.W. and P.S.; investigation, K.W., P.S., W.P., T.W., and S.S.; writing—original draft preparation, K.W. and P.S.; writing—review and editing, W.P., T.W., and S.S.; visualization, K.W. and P.S.; supervision, W.P. and T.W. All authors read and agreed to the published version of the manuscript.

**Funding:** This research received no external funding.

**Conflicts of Interest:** The authors declare no conflict of interest.

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
