# Peer review of "Torsional Stability Assessment of Columns Using Photometry and FEM"

_buildings, doi:10.3390/buildings10090162_

Round 1
Reviewer 1 Report
Comments
This paper investigated the torsional stability assessment of columns. The outcome is interesting for readers. However, there are several aspects that need to be improved. The reviewer can only recommend for publication if the author satisfactorily address the following comments in the revised version.
- This paper is based on numerical analysis which was not validated by the experimental investigation. How the author can verify the results obtained from this model?
- The justification for investigated parameters need to be presented.
- What the arrow indicates in Figure 9, and 10? The colour bar is needed in Fig 12.
- The conclusion of this paper need to be separated from the discussion section.
- How this study will contribute to the scientific knowledge that need to be mentioned at the last paragraph of introduction section.
- What boundary conditions and failure criteria were considered in finite element modelling?
- The introduction section has not provided sufficient information on the current approach of column strengthening. Mohammed et al. [Ref: Experimental and numerical evaluations on the behaviour of structures repaired using prefabricated FRP composites jacket] strengthened column using FRP jacket. Siddika et al. [Ref: Performances, challenges and opportunities in strengthening reinforced concrete structures by using FRPs-A state-of-the-art review] reviewed the strengthening of columns according to the current practice. This information can help to improve introduction section and suggest to include in the revised version.
I would be happy to see the revised version to understand how these comments are addressed.
Author Response
Point 1. This paper is based on numerical analysis which was not validated by the experimental investigation. How the author can verify the results obtained from this model?
Response 1: There is no possibility of verifying the load-bearing capacity of existing column in real construction using e.g. destructive test, because building is still in use. The aim of the article was to estimate load-carrying capacity of a reference column based on its real deformations read from point cloud.
It is possible to investigate that column by creating laboratory experiment with real scale or scaled down (due to its large dimensions) column, but it is out of the scope of this paper.
On the basis of numerous experiments (e.g. 1, 2, 3, 4), there is a significant convergence between the outcome from physical and numerical (based on FEM) experiments. Steel is a macroscopically homogeneous material and for simple schemes there is no need to perform physical experiments at each time.
(1) J. Barnat, M. Bajer, M. Vild, J. Melcher, M. Karmazínová, J. Piják, “Experimental Analysis of Lateral Torsional Buckling of Beams with Selected Cross-Section Types”, Procedia Engineering 195 (2017) 56 – 61
(2) J. Jankowska-Sandberg, J. KoÅ‚odziej, „Experimental study of steel truss lateral-torsional buckling”, Engineering Structures 46 (2013) 165–172
(3) M. Horacek, J. Melcher, O. Pesek, J. Brodniansky, “Focusing on Problem of Lateral Torsional Buckling of Beams with Web Holes”, Procedia Engineering 161 (2016) 549 – 555
(4) T. Tankova, J. P. Martins, L. S. da Silva, L. Marques, H. D. Craveiro, A. Santiago, “Experimental lateral-torsional buckling behaviour of web tapered I-section steel beams”, Engineering Structures 168 (2018) 355–370
Point 2. The justification for investigated parameters need to be presented.
Response 2: During the analysis of the cloud point the torsional deformations were noticed. Due to large load applied to the structure it was decided to estimate the load-carrying capacity of the reference column. Initial geometrical imperfection was used in the FEM analysis based on torsional deformations measured from the point cloud.
Point 3. What the arrow indicates in Figure 9, and 10? The colour bar is needed in Fig 12.
Response 3: In Figures 9 and 10 arrows indicates bolted connections. An explanation was added under those figure. The color bar was added in Fig. 12.
Point 4. The conclusion of this paper need to be separated from the discussion section.
Response 4: The names of paragraphs 5 and 6 were changed accordingly to its content.
Point 5. How this study will contribute to the scientific knowledge that need to be mentioned at the last paragraph of introduction section.
Response 5: Using point cloud data to obtain initial geometrical imperfection which is implemented in FEM calculations to estimate load-carrying capacity of steel elements is a new approach of describing the technical condition of structural elements using modern technologies.
The presented analysis is particularly useful in determining the load-bearing capacity of existing structures due to possibility of reading the actual geometry of deformations. 3D laser scanning may be especially useful when there are difficulties with accessibility, high altitude, temperature, etc.
Point 6. What boundary conditions and failure criteria were considered in finite element modelling?
Response 6: The boundary conditions were very important for defining structure properly. It was described in paragraph 4.1.
The analyses were conducted until a significant increase in the lateral displacement of the reference node as a function of the applied load was observed. It was interpreted as a loss of stability of structure leading to its destruction.
Point 7. The introduction section has not provided sufficient information on the current approach of column strengthening. Mohammed et al. [Ref: Experimental and numerical evaluations on the behaviour of structures repaired using prefabricated FRP composites jacket] strengthened column using FRP jacket. Siddika et al. [Ref: Performances, challenges and opportunities in strengthening reinforced concrete structures by using FRPs-A state-of-the-art review] reviewed the strengthening of columns according to the current practice. This information can help to improve introduction section and suggest to include in the revised version.
Response 7: The more information about column strengthening was added to the introduction section.
Reviewer 2 Report
The article focuses on a major problem that arises when dealing with an existing complex structure, that is the identification of its geometric characteristics, built conditions and of the current state. The combination of two techniques employed, the photometry and FEM, can substantialy improve the results the numerical analysis and allow for the design of suitable solutions to improve its structural behavior.
The aim of the research presented here is to propose a solution that reduces the lateral torsional bucking problems of the columns. The numerical methods used to asses the current state of the structure are the most used ones, namely the FEM, however in the literature there are other, numerically more efficient methods to tackle this problems, like generalized beam theory (GBT). However the interest here is to provide a more practical approach that is readily available to the practicing engineers.
Abstract: should be reformulated in order to state clearly the objectives of the paper, the methodology used and the results that were obtained. In the present version the abstract is somehow inconsistent.
Keywords: should be reviewed to reflect the content of the paper. For example LIDAR (Light Detection And Ranging) is not explicitly stated in the text nor referred.
Figures and equations must be referred in the text, eventually before they appear.
Example: Figure 3 on page 4 is not referred in the text. Captions should be reviewed in order to clearly indicate their content, e.g: “Figure 3. Explanation of equation (1)” would be better to change to: “Current and reference cross section of an open profile. “
In Line 118 “variable N” was previously defined as axial force and now called simply variable.
The derivation of the equation 2 is confusing as the description of the steps would require the splitting of the equation into two separate equations as the statement,“Following the substitution of Eq. 1 into Eq. 2, a system of homogeneous equations is created, where variable N can be determined if matrix determinant is such that A, B and C equal zero.” Lines 119 and 120 are not readily understandable
Line 123: Variable L doesn't seem to appear in equation (2). It appears later in equation (3).
Line 182: Figure 4 should be referred in the text and its quality needs be improved.
Line 215 states that “It is a spatial model that accounts non geometric and material non linearity” It is better to state as a 3D model instead of “spatial model” and “non” it is probably means “for”?
The weakness of the numerical model is that the values of the material properties are not given explicitly, moreover Line 218 and 219 a reference is given ([7]) turns out being a non-English document and probably not publicly available, thus hindering the reproduciblity of the results.
Is it possible the over the time the columns suffer from strength degradation? Is it possible to identify by leaser scanning? Would it have any importance on the results?
Figure 5 it is better to be placed following Line 223.
Figure 9(c) the arrows are pointing erroneously.
Figure 13 – The caption should mention the scale factor used to represent the imperfections.
Section 5: Results – must be carefully reviewed in order to maintain the same writing style throughout the paper.
Author Response
Abstract: should be reformulated in order to state clearly the objectives of the paper, the methodology used and the results that were obtained. In the present version the abstract is somehow inconsistent.
Response: Abstract was rewritten to underline the objectives, used methodology and obtained results.
Keywords: should be reviewed to reflect the content of the paper. For example LIDAR (Light Detection And Ranging) is not explicitly stated in the text nor referred.
Response: Keywords were reviewed to adjust them to the content of the paper.
Figures and equations must be referred in the text, eventually before they appear.
Response: References of figures and equations were added to the text (where missing).
Example: Figure 3 on page 4 is not referred in the text. Captions should be reviewed in order to clearly indicate their content, e.g: “Figure 3. Explanation of equation (1)” would be better to change to: “Current and reference cross section of an open profile. “
Response: Caption under Figure 3 was changed as suggested.
In Line 118 “variable N” was previously defined as axial force and now called simply variable. The derivation of the equation 2 is confusing as the description of the steps would require the splitting of the equation into two separate equations as the statement, “Following the substitution of Eq. 1 into Eq. 2, a system of homogeneous equations is created, where variable N can be determined if matrix determinant is such that A, B and C equal zero.” Lines 119 and 120 are not readily understandable.
Response: Equation (1) is between lines 118-119. The first part of equation (1) was deleted due to clarify the text. Explanation of equation (2) was corrected. Deriving critical forces of bending and twisting is out of scope of the paper and clarification of functions with coefficients A, B and C would unnecessarily expand the article.
Line 123: Variable L doesn't seem to appear in equation (2). It appears later in equation (3).
Response: It was corrected – definition of L appeared under equation (3).
Line 182: Figure 4 should be referred in the text and its quality needs be improved.
Response: Quality of Figure 3 and Figure 4 was improved. In Figure 4 the plate stiffeners were marked as grey.
Line 215 states that “It is a spatial model that accounts non geometric and material non linearity” It is better to state as a 3D model instead of “spatial model” and “non” it is probably means “for”?
Response: “Non” was changed to “for” – it was a linguistic mistake. “Spatial model” was changed to “3D model” as suggested.
The weakness of the numerical model is that the values of the material properties are not given explicitly, moreover Line 218 and 219 a reference is given ([7]) turns out being a non-English document and probably not publicly available, thus hindering the reproduciblity of the results.
Response: Steel properties were described in chapter 2.1 (yield strength and the class of steel – line 75-78). Material constituents were described in chapter 4.1 (line 222-224). Reference [7] (in the actual version [13]) is a decommissioned Polish code and it is available in the Polish Standardization Committee website. The properties of steel St3S can be easily find in the other scientific papers or so.
Is it possible the over the time the columns suffer from strength degradation? Is it possible to identify by leaser scanning? Would it have any importance on the results?
Response: There is no possibility of strength degradation estimation using laser scanning. The measurement delivers information concerning deformations of the column geometry. Only if there will be a large loss of material in some part of the element due to corrosion it would be noticeable.
According to the authors opinion, the strength of the steel didn’t degraded significantly through the building’s “life”. There was no big changes of temperature and the columns was indoor, so even the hardening effect isn’t crucial.
Figure 5 it is better to be placed following Line 223.
Response: It was placed as suggested.
Figure 9(c) the arrows are pointing erroneously.
Response: The arrows were corrected.
Figure 13 – The caption should mention the scale factor used to represent the imperfections.
Response: Information about scale factor was added in the caption under Figure 13.
Section 5: Results – must be carefully reviewed in order to maintain the same writing style throughout the paper.
Response: Writing style in section 5 was reviewed and corrected.
Reviewer 3 Report
The manuscript presented the torsional buckling analysis of a steel open section column using shell models. The frame structure was modeled based on the scanned points of cloud of a real structure. It was revealed that how to mount the spandrel beams directly affects the damage mechanism and failure pattern. A method for improving the ultimate load-carrying capacity was also proposed. The manuscript is very well written, and the topic falls within the scope of the journal. Before recommending publication, I would like to ask the authors to address the remarks listed below.
(1) This comment may fall out of the scope of the paper, but it would be interesting to see if the previous strengthening work improved the stability of the structure in terms of torsional buckling by comparing the renovated structure and the original one based on design.
(2) Can the authors briefly talk about the process of turning the scanned points of cloud into a FE model? I believe the detailed approach will be interesting to a lot of readers.
(3) The authors wrote in section 4.1 that the model accounts non geometric and material nonlinearity, which is a bit misleading. I thought both geometric and material nonlinearities have to be considered for instability analysis.
(4) The element types that are used in the Abaqus model are suggested to be mentioned. For example, I assume S4R was used for the modeling of flanges and webs. Also, why not S4 (i.e., full integration)?
(5) In Figures 12, 13, 14, and 18, the authors are suggested to turn off the mesh display to better visualize the configuration. The mesh makes a lot of regions dark and hard to see.
(6) The authors are suggested to include a few recent publications on the buckling and damage analysis of structures: (1) doi.org/10.1016/j.ijnonlinmec.2017.06.003 (2) doi.org/10.1061/(ASCE)EM.1943-7889.0001263 (3) doi.org/10.1016/j.jcsr.2016.08.013 (4) doi.org/10.1016/j.compstruc.2017.10.016
Author Response
Point 1. This comment may fall out of the scope of the paper, but it would be interesting to see if the previous strengthening work improved the stability of the structure in terms of torsional buckling by comparing the renovated structure and the original one based on design.
Response 1: In the considered part of the power plant, the columns had not been strengthened before. Therefore authors did not perform such analyses. Perhaps such analyses will be conducted in the future.
Point 2. Can the authors briefly talk about the process of turning the scanned points of cloud into a FE model? I believe the detailed approach will be interesting to a lot of readers.
Response 2: The geometry from the point cloud was not directly imported into FEM model. Based on the point cloud, the element geometry was modelled in ABAQUS, it was not an automatic process. The information obtained from the point cloud also allowed to determine the size and shape of the geometric imperfections included in the FEM model.
Point 3. The authors wrote in section 4.1 that the model accounts non geometric and material nonlinearity, which is a bit misleading. I thought both geometric and material nonlinearities have to be considered for instability analysis.
Response 3: It was a linguistic mistake and it was corrected. Geometric and material nonlinearities were considered.
Point 4. The element types that are used in the Abaqus model are suggested to be mentioned. For example, I assume S4R was used for the modeling of flanges and webs. Also, why not S4 (i.e., full integration)?
Response 4: The model was recalculated using shell elements with full integration and linear shape function (S4) and there was no significant difference between outcomes, but the calculations last longer. Therefore, S4R element were adopted for further analyses. The explanation was added in lines 219 – 222.
Point 5. In Figures 12, 13, 14, and 18, the authors are suggested to turn off the mesh display to better visualize the configuration. The mesh makes a lot of regions dark and hard to see.
Response 5: Figures 12, 13, 14 and 18 were corrected.
Point 6. The authors are suggested to include a few recent publications on the buckling and damage analysis of structures: (1) doi.org/10.1016/j.ijnonlinmec.2017.06.003 (2) doi.org/10.1061/(ASCE)EM.1943-7889.0001263 (3) doi.org/10.1016/j.jcsr.2016.08.013 (4) doi.org/10.1016/j.compstruc.2017.10.016
Response 6: Suggested publications were added and commented in the introduction (chapter 1).
Round 2
Reviewer 1 Report
I have no further comments.